# Deep Analysis of Clinical Parameters and Temporal Evolution of Glycemic Parameters Based on CGM Data for the Characterization of Severe Hypoglycemia in a Cohort of Children and Adolescents with Type 1 Diabetes

**DOI:** 10.3390/nu15132957

**Published:** 2023-06-29

**Authors:** Antoine Harvengt, Maude Beckers, Laure Boutsen, Elise Costenoble, Chloé Brunelle, Philippe Lysy

**Affiliations:** 1Pôle PEDI, Institut de Recherche Expérimentale et Clinique, UCLouvain, 1200 Brussels, Belgium; antoine.harvengt@uclouvain.be; 2Specialized Pediatrics Service, Cliniques Universitaires Saint-Luc, 1200 Brussels, Belgium; maude.beckers@saintluc.uclouvain.be (M.B.); laure.boutsen@saintluc.uclouvain.be (L.B.); elise.costenoble@saintluc.uclouvain.be (E.C.); chloe.brunelle@saintluc.uclouvain.be (C.B.)

**Keywords:** children, clinical markers, continuous glucose monitoring, glycemic variability, hypoglycemia, insulin dose-adjusted A1C, type 1 diabetes, severe hypoglycemia, target, therapeutic management

## Abstract

This study aims to evaluate the determinants and clinical markers of patients at risk for severe hypoglycemia (SH) in children and adolescents with type 1 diabetes. In the EPI-GLUREDIA study, clinical parameters and continuous glucose monitoring metrics from children and adolescents with type 1 diabetes were retrospectively analyzed between July 2017 and June 2022. Their clinical parameters were collected during traditional and quarterly medical consultations according to whether they experienced severe hypoglycemia or not. Then, continuous glucose monitoring metrics were analyzed on days surrounding SH during specific periods. According to the glycemic parameters, glycemic hemoglobin and glycemic mean were significantly lower in the three months preceding a SH compared with during three normal months (*p* < 0.05). Moreover, the time spent in hypoglycemia(time below the range, TBR_<3.3_) and its strong correlation (R = 0.9, *p* < 0.001) with the frequency of SH represent a sensitive and specific clinical parameter to predict SH (cut-off: 9%, sensitivity: 71%, specificity: 63%). The second finding of the GLUREDIA study is that SH is not an isolated event in the glycemic follow-up of our T1DM patients. Indeed, most of the glycemic parameters (i.e., glycemic mean, glycemic variability, frequency of hypoglycemia, and glycemic targets) vary considerably in the month preceding an SH (all *p* < 0.05), whereas most of these studied glycemic parameters remain stable in the absence of a severe acute complication (all *p* > 0.05). Furthermore, the use of ROC curves allowed us to determine for each glycemic parameter a sensitive or specific threshold capable of more accurately predicting SH. For example, a 10% increase in the frequency of hypoglycemia predicts a risk of near SH with good combination of sensitivity and specificity (sensitivity: 80%, specificity: 60%). The GLUREDIA study aimed to target clinical and glycemic parameters to predict patients at risk for SH. First, we identified TBR_<3.3_ < 9% as a sensitive and specific tool to reduce the frequency of SH. In addition, SH was not an isolated event but rather it was accompanied by glycemic disturbances in the 30 days before SH.

## 1. Introduction

Type 1 diabetes mellitus (T1D) is characterized by chronic hyperglycemia and insulinopenia consecutive to an autoimmune reaction that massively reduces the function of pancreatic β cells. Shortly after the initiation of insulin therapy, a majority of patients enter partial remission, a transient period defined by an increase in residual β-cell insulin secretion that clinically translates in both reduced daily insulin requirements and improved glycemic control. After partial remission, a progressive increase in exogenous insulin dependence leads to a rise in glycemic variability and difficulties in avoiding hypoglycemia [1]. Even under conventional intensive insulin therapy, a high frequency of hypoglycemia (estimated at 5–15% of total blood glucose measurements) is necessary to maintain the glycemic mean within treatment targets (i.e., <8.25 mmol/L) [2,3].

Hypoglycemia, characterized by low levels of blood (i.e., plasma or capillary) or subcutaneous glucose (i.e., <3.3 mmol/L), is thus the most frequent acute complication of T1D and results from an imbalance between insulin requirements, insulin dose, and carbohydrate intake [4]. Depending on the severity of related symptoms, hypoglycemia is graded from mild to severe, with severe hypoglycemia (SH) corresponding to any blood glucose level below 2.97 mmol/L accompanied by an alteration in patient consciousness [5].

Symptomatic hypoglycemia is responsible for reduced quality of life in children with T1D and their families. Several studies have highlighted that SH fosters symptoms of depression and anxiety in a significant proportion of patients with T1D and their parents. Parents may also be worried about potential acute and long-term complications of hypoglycemic events [6,7]. Moreover, an episode of SH can trigger a psychological and physiological barrier to optimal diabetes management in the weeks following an episode [3]. Unfortunately, SH remains sudden for most patients and for the diabetes care team, especially since no markers or tools exist that can effectively predict the occurrence of SH in patients with T1D.

For many years, studies have shown that glycated hemoglobin (HbA_1C_) and glycemic mean were significantly lower in patients with the highest risk of SH [8]. However, several recent longitudinal studies have shown a decrease in the frequency of SH in parallel with a reduction in the mean level of HbA_1C_, partly related to an improvement in the overall therapeutic management of diabetes [9,10,11,12], leaving us without usable markers to identify patients at risk of SH. The objective of this analysis is to identify glycemic and clinical markers of risk for SH in children and adolescents with type 1 diabetes.

## 2. Methodology

### 2.1. Study Design and Participants

The Epi-GLUREDIA study, a subsidiary part of the GLUREDIA consortium study, is a retrospective, monocentric clinical study designed to characterize SH in children and adolescents with type 1 diabetes. The study took place entirely in the pediatric diabetes department of Cliniques universitaires Saint-Luc (CUSL, Brussels, Belgium). All data were collected and analyzed in accordance with the ethical Declaration of Helsinki. The GLUREDIA study was approved by the Hospital-Faculty Ethics Committee of the CUSL (2022/02/FEV/043). Patient enrollment began on 1 July 2017 and ended on 30 June 2022.

Patients eligible to participate in the Epi-GLUREDIA study were 2 to 20 years old and were diagnosed with type 1 diabetes according to American Diabetes Association (ADA) criteria, such as fasting blood glucose ≥ 6.93 mmol/L and/or blood glucose ≥ 11 mmol/L at the 120th minute of OGTT; HbA_1C_ ≥ 6.5%; and/or symptoms of hyperglycemia/hyperglycemic crisis with random glucose ≥ 11 mmol/L and presence of one or more anti-islet autoantibodies (i.e., anti-insulin, anti-IA2, anti-GAD65, and anti-ZnT8) in the serum [1,13].

Exclusion criteria for the study were age <2 years or >20 years; absence of anti-islet autoantibodies; use of drugs that may interfere with insulin secretion or sensitivity (i.e., sulfonylureas, diazoxide, somatostatin, methylxanthic derivatives, corticosteroids, biguanides, and incretins); the presence, at inclusion, of celiac disease diagnosed by a pathological duodenal biopsy within one month; the presence, at inclusion, of an autoimmune or autoinflammatory disease other than T1D or of an active malignant disease; morbid obesity defined by a body mass index (BMI) with a Z-score greater than three standard deviations; hepatic, renal, or adrenal insufficiency; dysmorphia with suspected underlying genetic syndrome; a history of bone marrow allograft or post-hemolytic-uremic syndrome diabetes; or participation in another study within the previous three months, with administration of blood derivatives or potentially immunomodulatory treatments.

### 2.2. Study Procedure

The Epi-GLUREDIA retrospective study consisted of an analysis of the clinical and glycemic parameters encoded in the medical records (EPIC^®^ software, © 1979–2022 Epic System Corporation, Verona, WI, USA) of patients currently enrolled in the CUSL pediatric diabetes convention of care. Clinical data (clinical and glycemic parameters) were recorded at each report of clinical follow-up visit of all our patients.

### 2.3. Severe Hypoglycemia

Severe hypoglycemia is characterized by an abnormally low blood glucose level, below 2.97 mmol/L. It manifests itself as altered consciousness in patients with diabetes [2]. The diagnosis is made by a diabetologist during consultation following an episode of severe hypoglycemia. Blood glucose levels and altered consciousness are assessed to confirm the condition.

### 2.4. Partial Remission

Remission status was determined using the IDAA_1C_ score, as follows: HbA_1c_ (%) + 4 × insulin dose (U/kg body weight per 24 h). The remission status is defined by a IDAA_1C_ score below 9 [14].

### 2.5. Continuous Glucose Monitoring

Raw continuous glucose monitoring metrics from a 90-day interval were extracted at each outpatient clinical visit from various continuous glucose monitoring devices (i.e., Freestyle Libre^®^, Abbott; Dexcom^®^, Dexcom; Enlite^TM^, Medtronic MiniMed). Raw glycemic data were pre-processed using the R (R Core Team (2021)) statistical package cgmanalysis. Hypoglycemia was defined as a value below 60 mg/dL (3.3 mmol/L), and hyperglycemia was defined as a value above 160 mg/dL (8.8 mmol/L), for all glycemic measurements (i.e., capillary and subcutaneous). Glycemic values below 20 mg/dL (1.1 mmol/L) or contiguous values with a dynamic change >100 mg/dL (5.6 mmol/L) in less than 5 min were considered as artifacts and were excluded from the analysis. Among the glycemic parameters studied were the proportion of time spent affected by hypoglycemia (time below the range; TBR_<3.3_), the proportion of time spent affected by normoglycemia (time in range; TIR_3.3–8.8_), and proportion of time spent affected by hyperglycemia (time above the range; TAR_>8.8_).

### 2.6. Statistical Analyses

All statistical analyses were performed R (R Core Team (2021). R: A language and environment for statistical computing. R Foundation for Statistical Computing, Vienna, Austria. URL [https://www.R-project.org/, accessed on 20 December 2022]). The level of statistical significance used for all analyzes was 0.05. Demographic and clinical data are reported as mean ± standard deviation for continuous variables and as numbers and proportions for categorical variables. Linear regression was used for continuous variables. The comparisons between groups were made using Student *t*-test (if two groups) or their non-parametric equivalent (Mann–Whitney U test), as appropriate. The comparisons between proportions were made using Chi-square test. The correlations were estimated by Pearson’s correlation coefficient. *p*-values were adjusted for multiple testing with the Bonferroni procedure [15].

## 3. Results

In our pediatric cohort of patients with T1D, we selected 441 medical outpatient visits from 356 pediatric patients occurring between July 2017 and June 2022, and divided these visits in two groups: one (severe hypoglycemia group or SH group) composed of all consultations (*n* = 162) that followed a recent SH episode (less than 3 weeks between SH and consultation; mean = 17 days), whereas the other (i.e., non-SH group) was composed of 279 random consultations not preceded by a severe acute complication (SH or ketoacidosis) in the 12 months preceding the consultation (flow chart; Figure 1).

First, we wanted to evaluate clinical and diabetes parameters that might distinguish SH and non-SH groups (Table 1). Although there was no difference in gender distribution and diabetes duration between the two groups (all *p* > 0.05), girls in the SH were slightly younger than girls in the non-SH. The proportion of girls treated with insulin pumps in the SH was significantly lower than controls. These differences were not observed in boys. Moreover, no significant difference existed between the two groups of patients regarding the frequency and average duration of partial remission, or the proportion of patients currently in partial remission.

Regarding glycemic parameters, we observed a significantly lower glycemic mean and HbA_1C_ in patients from the SH compared with the non-SH (Table 1). Furthermore, HbA_1C_ correlated intensely but inversely with the proportion of consultations preceded by SH (number of consultations SH/total number of consultations according to the value of HbA_1C_ rounded to the decimal point) (R = −0.87; *p* < 0.001). In opposition, time spent affected by hypoglycemia (time below the range < 3.3 mmol/L; TBR_<3.3_) was significantly greater for patients in the SH (vs. non-SH). Additionally, we demonstrated an intense positive correlation between TBR_<3.3_ and the proportion of visits preceded by SH (R = 0.9; *p* < 0.001). Next, we performed a preliminary logistic regression to determine the most reliable parameter for predicting severe hypoglycemia. The first three parameters considered were mean blood glucose (Pr < 0.001), glycated hemoglobin (Pr < 0.001), and target glucose level (Pr < 0.001). Finally, the ROC curve analysis showed a TBR_<60_ at 9% to be the value with the best combination of sensitivity and specificity for determining a risk of SH (sensitivity: 71%, specificity: 63%, relative risk: 2.9) (Figure 2).

In a second step, we were able to collect and analyze 42 CGM records having recorded a SH event. The glycemic parameters analyzed were glycemic mean, coefficient of glycemic variability (CV), frequency of hypoglycemia, mean duration of hypoglycemia, time spent affected by normoglycemia (time in range (3.3; 8.8 mmol/L); TIR_3.3–8.8_), time spent affected by hypoglycemia (time below the range (<3.3 mmol/L); TBR_<3.3_), and time spent affected by hyperglycemia (time above the range (>8.8 mmol/L); TAR_>8.8_).

First, we analyzed the temporal evolution of glycemic parameters during the five days preceding SH. Specifically, we assessed whether the different glycemic parameters varied daily according to a specific pattern during the 5 days preceding SH (Figure 3A). While there were significant inter-day variations for the majority of glycemic parameters studied (*p* < 0.05), we observed no specific pattern of temporal evolution of the different glycemic parameters studied during these five days (*p* > 0.05). Indeed, no glycemic parameter remained stable during the five days preceding SH, yet the daily modification of these parameters was not uniform for all patients (Figure 4).

Next, we compared the glycemic parameters between a period of five days directly preceding a SH (X1) and a period of five days (X2) more than 30 days away from any acute severe complication (SH or diabetic ketoacidosis) (Figure 3B). First, we observed a significantly lower TAR_>8.8_ and glycemic mean on the five days prior to SH compared with the five days away from any severe acute complication (all *p* < 0.05). In contrast, CV, hypoglycemia frequency, mean duration of hypoglycemia, and TBR_<3.3_ were significantly greater during X1 compared with the X2 period (All *p* < 0.05). Finally, we observed no significant difference for TIR_3.3–8.8_ between the two periods (*p* > 0.05) (Figure 5).

Subsequently, we studied the evolution of glycemic parameters during a 60-day period that starts 30 days before SH and ends 30 days after SH. For this purpose, we divided this timing into four distinct parts of 5 days: H1, a period that started 30 days before SH; H2, a period that directly preceded SH; H3, a period that directly followed SH; and H4, a period that ended 30 days after SH (Figure 3C). In this way, we observed that all glycemic parameters evolved differently and significantly frequently between each of these periods (Figure 6). For example, glycemic mean and TAR_>160_ decreased between H1 and H2 (*p* < 0.05), remained stable between H2 and H3 (*p* > 0.05), and then increased between H3 and H4 (*p* > 0.05). On the other hand, the CV did not significantly change across the four periods (*p* > 0.05). Finally, the frequency of hypoglycemia reached its maximum during H2 and then decreased.

Finally, we compared these glycemic parameters between two periods (P1–P2) of 5 days separated by 20 days without any severe acute complication related to T1D (Figure 3D). The objective was to determine how glycemic parameters naturally varied over time, outside of severe events. Most glycemic parameters remained stable over time (P1 vs. P2, all *p* > 0.05). Only TIR_3.3–8.8_ and TBR_<3.3_ varied (All *p* < 0.05) between these two uncomplicated periods (Figure 7).

Finally, these differences in the variation in glycemic parameters between two uncomplicated periods (P1–P2) and a group of periods preceding SH (H1–H2) encouraged us to calculate the intensity of the variation in glycemic parameters between these two periods separated by 20 days. The variation in glycemic parameters was calculated on CGM data either containing SH (H1–H2 periods) or including no severe acute diabetes-related complications (P1–P2 periods).These periods have been described above. Finally, the intensity of the variation in the glycemic parameters was calculated using the following formula ([H2–H1]/H1) or ([P2–P1]/P1). With these calculated data, we used ROC curves to determine the thresholds of variation in all glycemic parameters to predict a risk of SH. For each parameter studied (i.e., glycemic mean, frequency of hypoglycemia, TAR_>8.8_, and TBR_<3.3_), we identified cut-offs that occurred with the best combination of sensitivity and specificity. Above the absolute values of these cut-offs, we observed a significant increased risk of SH. For example, an increase of more than 50% in TAR_>160_ is associated with an increased risk of SH (sensitivity: 75%, specificity: 53%). Furthermore, we found that variation in the frequency of hypoglycemia (threshold: 10%, sensitivity: 80%, specificity: 60%) offers the best combination of sensitivity and specificity for predicting the risk of SH (Table 2).

## 4. Discussion

The first message of our study is the absence of a specific profile of patients particularly at risk of SH based solely on clinical parameters. Indeed, in our investigation, the duration of the disease, the age of the patient, and the gender of the patient did not have any major impacts on the frequency of SH. In other words, SH appeared to be an acute complication that can occur at any time in all our diabetic patients. Only pubertal girls seemed to be partially protected against SH, compared with boys. This observation could be explained, in part, by a decrease in insulin sensitivity related to sex hormones [16]. Moreover, while the insulin pump generally allows for a better glycemic balance and a better control of diabetes [17,18], this treatment seems to partially protect only girls from SH and not boys. Therefore, it seems essential that all patients with de novo diabetes should be adequately and regularly educated by the medical team at diagnosis to recognize and to better treat SH [1,2,19].

Additionally, our study shows that patients in partial remission were not protected from SH. Moreover, the duration or the frequency of partial remission did not protect patients from SH in the long term. Therefore, partial residual insulin secretion does not seem to play a major direct role in the pathophysiology of SH.

While the GLUREDIA study did not find any clinical parameters that effectively predicted SH, there were significant differences in the glycemic parameters between the SH and the non-SH. Although some parameters (i.e., mean glycemia and HbA_1C_) were previously discussed in smaller-cohort or older clinical studies [8,12], we identified new glycemic parameters to distinguish patients at risk of SH. To select the appropriate clinical parameter, we performed a preliminary logistic regression to determine the most reliable parameter for predicting severe hypoglycemia. The first three parameters considered were mean blood glucose, HbA_1C_, and TBR_<3.3_. However, the first two parameters occurred with a major drawback. Increasing mean blood glucose and HbA_1C_ could reduce the risk of severe hypoglycemia, but it could also increase the risk of long-term chronic complications. We therefore chose to focus on reducing TBR_<3.3_.Time spent affected by hypoglycemia and its strong correlation with the frequency of SH represented a sensitive and specific clinical parameter to predict SH. Therefore, it may be important to revise the management of patients with T1D [20,21,22] and aim for a TBR_<3.3_ below 9% at each medical visit. Having a TBR_<3.3_ below 9% would reduce the risk of SH by 60%.

The clinical and glycemic parameters collected at each medical consultation offered us a new therapeutic target to consider when planning therapeutic management and to reduce the risk of SH. However, as mentioned above, SH is always an unexpected event for the patient and their relatives.

The second message of the GLUREDIA study is that SH is not an isolated event in the glycemic follow-up of our patients with T1D. Indeed, most glycemic parameters vary considerably in the thirty days preceding a SH, whereas most of these studied glycemic parameters remained stable in the absence of a severe acute complication. As most of the glycemic parameters vary significantly before SH while they remain stable in the absence of severe acute complications, the great advantage of these observations is that it would be possible to determine the exact moment of occurrence of hypoglycemia according to the intensity of variation in these glycemic parameters.

Thus, with the help of ROC curves, we determined for each parameter a sensitive or specific cut-off point capable of more accurately predicting SH. Consequently, the calculated sensitive and specific thresholds of variation in theses glycemic parameters could become new tools to prevent the risk of a near occurrence of SH. For example, a 10% increase in the frequency of hypoglycemia predicts a risk of near SH with a good combination of sensitivity and specificity (Se: 80%, Sp: 60%). To our knowledge, these observations have not yet been described in the literature and they could allow for improvements in the quality of life of the patients in the future based on their predictive characteristics.

## 5. Conclusions

The GLUREDIA study aimed to target clinical and glycemic parameters to predict patients at risk of SH. While no clinical parameters were found to predict the risk of SH, we identified TBR < 9% as a sensitive and specific tool to reduce the frequency of SH. Furthermore, when analyzing CGM records, our GLUREDIA study demonstrated that SH was not an isolated event but was accompanied by glycemic disturbances in the 30 days preceding SH. In addition, thresholds of variation in the calculated glycemic parameters could be a new tool to predict a SH event quite accurately.

## 6. Limitations

One of the main limitations of the GLUREDIA study is caused by the small amount of glycemic data (42 CGMs) available in the medical records of the patients recruited for the study. A second limitation is that GLUREDIA is a single-center and not a multicenter study.

## 7. Perspectives

A first possibility is to analyze a composite of biological parameters to better target patients at risk of SH. Another possibility is to add additional glycemic parameters in the patient medical records to compare which patient is at risk of SH.

## Figures and Tables

**Figure 1 nutrients-15-02957-f001:**
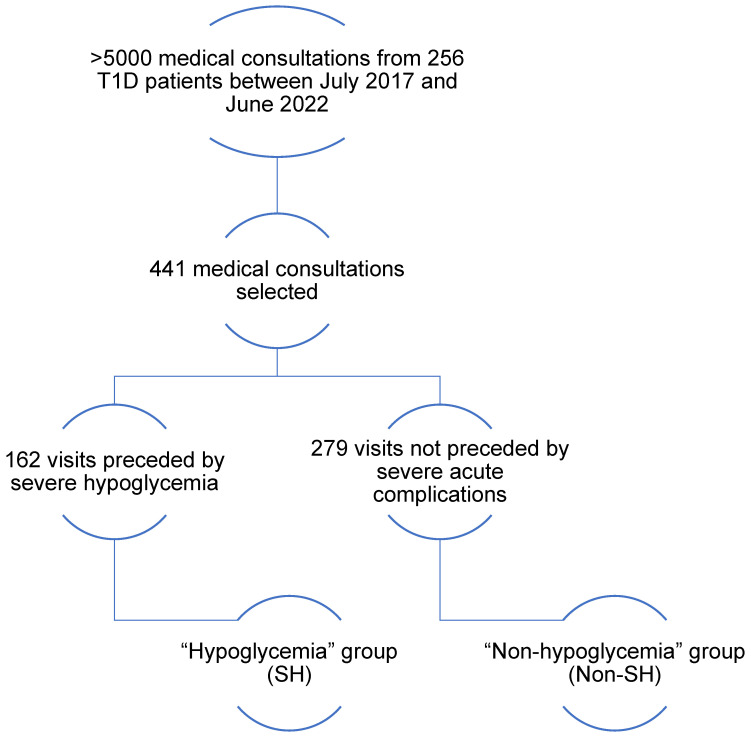
Flowchart. Between July 2017 and June 2022, there were over 5000 pediatric diabetes visits; 441 medical consultations were selected from the 5000: 162 consultations were preceded by severe hypoglycemia in the past 3 weeks; these 162 consultations constitute SH; 279 consultations were randomly selected and not preceded by severe hypoglycemia in the previous 12 months; these 279 consultations represent the non-SH.

**Figure 2 nutrients-15-02957-f002:**
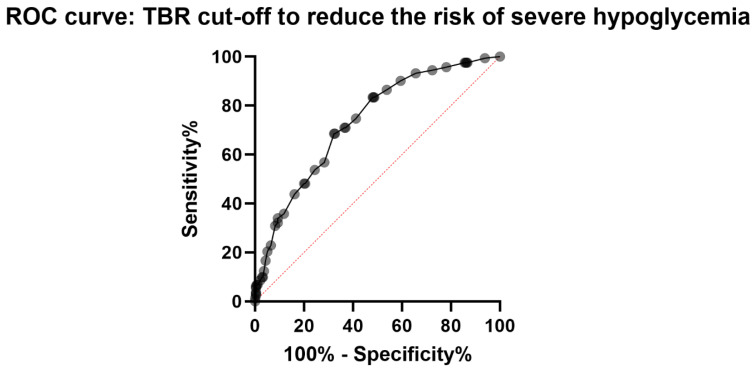
Representation of the ROC curves used to determine a TBR cut-off to reduce the risk of occurring severe hypoglycemia.

**Figure 3 nutrients-15-02957-f003:**
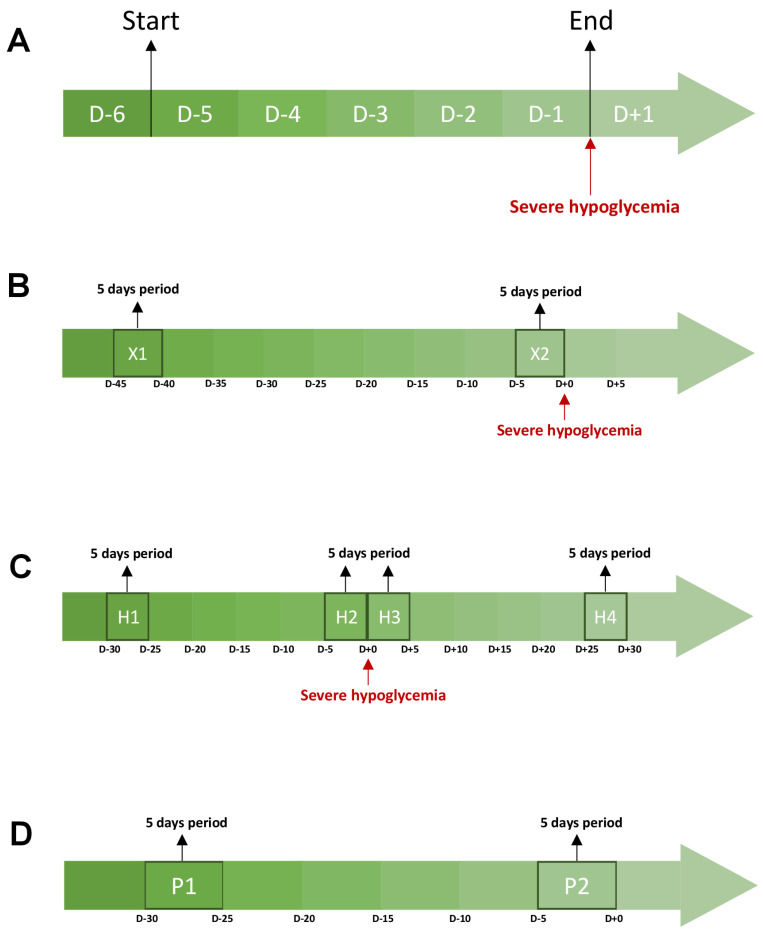
Description of the periods studied during the CGMs analysis. Each graph represents a cgms analysis. The orange areas correspond to the periods studied. (**A**) represents the evolution of glycemic parameters during the 5 days preceding a severe hypoglycemia. (**B**) compares the glycemic parameters between a 5-day period directly preceding a severe hypoglycemia and a 5-day period more than 30 days away from any acute severe complication (ketoacidosis and severe hypoglycemia). (**C**) studies the evolution of glycemic parameters between four periods around a severe hypoglycemia. H1 is a 5-day period starting 30 days before severe hypoglycemia. H2 is a 5-day period directly preceding a severe hypoglycemia. H3 is a 5-day period directly following a severe hypoglycemia. H4 a 5-day period that ends 30 days after a severe hypoglycemia. (**D**) studies the evolution of glycemic parameters between two periods of 5 days and 20 days apart from any acute severe complication (ketoacidosis and severe hypoglycemia). Abbreviations: D-(number), number of days before severe hypoglycemia (**A**–**C**) or the end of the evaluation of glycemic parameters (**D**).

**Figure 4 nutrients-15-02957-f004:**
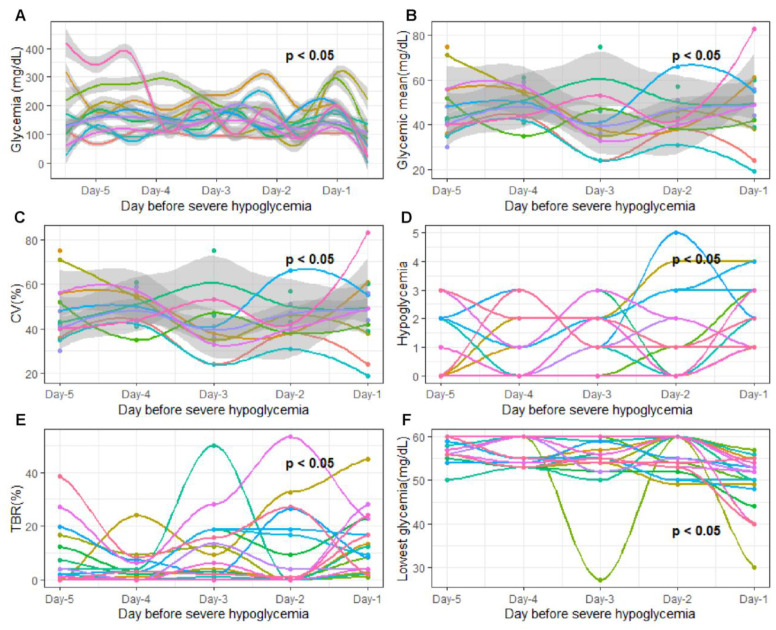
Evolution of glycemic parameters 5 days before severe hypoglycemia. The x-axis represents the 5 days studied before severe hypoglycemia. Day-5, 5 days before severe hypoglycemia; Day-4, 4 days before severe hypoglycemia; Day-3, 3 days before severe hypoglycemia; Day-2, 2 days before severe hypoglycemia; Day-1; 24 h before severe hypoglycemia The y axis corresponds to the glycemic parameters: (**A**) glycemia (mg/dL); (**B**) daily glycemic mean (mg/dL); (**C**) glycemic variability; (**D**) hypoglycemia frequency (n/day); (**E**) time spent affected by hypoglycemia (<3.3 mmol/L); (**F**) lowest recorded glycemic value (mg/dL). The curve on the graph models the variation in the studied glycemic parameters via polynomial regression. The gray areas following the blue curve represent the confidence interval of the polynomial regression. Each color of the graph corresponds to a single patient. On all graphs, *p*-values are written. Each *p*-value determines the variation in each glycemic parameter as a specific temporal evolution pattern. Temporal evolution pattern was considered as significant when *p*-value was under 0.05. *p*-values were adjusted for multiple testing with the Bonferroni procedure after a paired test.

**Figure 5 nutrients-15-02957-f005:**
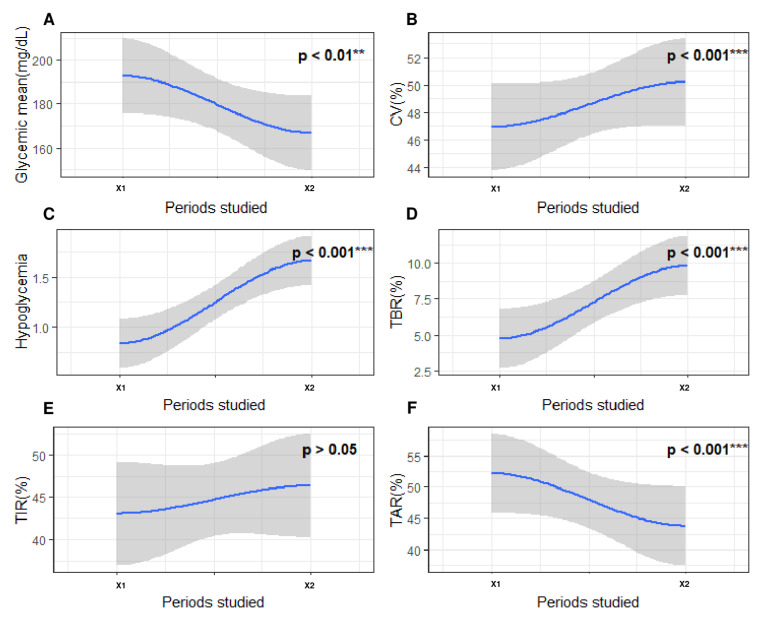
Evolution of glycemic parameters between a period of five days directly preceding a SH episode and a period of five days more than 30 days away from any acute severe complication (SH or diabetic ketoacidosis). The x-axis represents the 2 periods studied around severe hypoglycemia. X1, a 5-day period more than 30 days away from severe hypoglycemia; X2, a 5-day period directly preceding severe hypoglycemia. The y axis corresponds to the glycemic parameters: (**A**) glycemic mean (mg/dL); (**B**) coefficient of variability (%); (**C**) frequency of hypoglycemia (n/day); (**D**) time spent affected by hypoglycemia (<3.3 mmol/L; %); (**E**) time spent affected by normoglycemia (3.3–8.8 mmol/L; %); (**F**) time spent affected by hyperglycemia (>8.8 mmol/L; %). The blue curve on the graph models the variation in the studied glycemic parameters via polynomial regression. The gray areas following the blue curve represent the confidence interval of the polynomial regression. On all graphs, *p*-values are written. Each *p*-value represents the statistical significance of the variation in the studied blood glucose parameter between the two periods concerned. Differences between two periods were considered as significant when *p*-value was under 0.05. The level of significance is represented as follows: nonsignificant (), *p* < 0.01 (**), *p* < 0.001 (***).

**Figure 6 nutrients-15-02957-f006:**
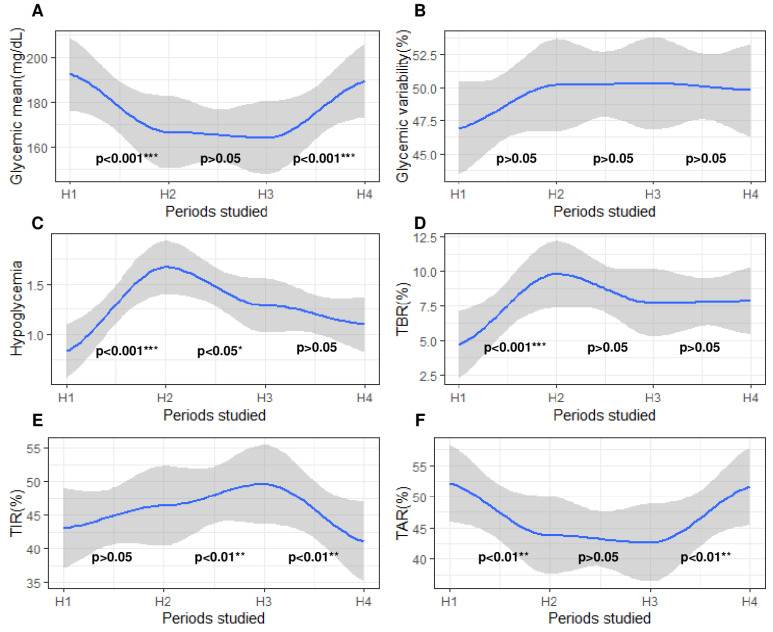
Evolution of glycemic parameters 30 days before and 30 days after severe hypoglycemia. The *x*-axis represents the 4 periods studied around severe hypoglycemia. H1, a 5-day period beginning 30 days before severe hypoglycemia; H2, a 5-day period directly preceding severe hypoglycemia; H3, a 5-day period directly following severe hypoglycemia; H4, a 5-day period ending 30 days after severe hypoglycemia. The *y*-axis corresponds to the glycemic parameters: (**A**) glycemic mean; (**B**) coefficient of variability; (**C**) frequency of hypoglycemia; (**D**) time spent affected by hypoglycemia (<3.3 mmol/L); (**E**) time spent affected by normoglycemia (3.3–8.8 mmol/L); (**F**) time spent affected by hyperglycemia (>8.8 mmol/L). On all graphs, 3 *p*-values are written, each located between two periods. The blue curve on the graph models the variation in the studied glycemic parameters by polynomial regression. The gray areas following the blue curve represent the confidence interval of the polynomial regression. Each *p*-value represents the statistical significance of the variation in the studied blood glucose parameter between the two periods concerned. Differences between two periods were considered as significant when *p*-value was under 0.05. *p*-values were adjusted for multiple testing with the Bonferroni procedure after a paired test. The level of significance is represented as follows: nonsignificant (), *p* < 0.05 (*), *p* < 0.01 (**), *p* < 0.001 (***).

**Figure 7 nutrients-15-02957-f007:**
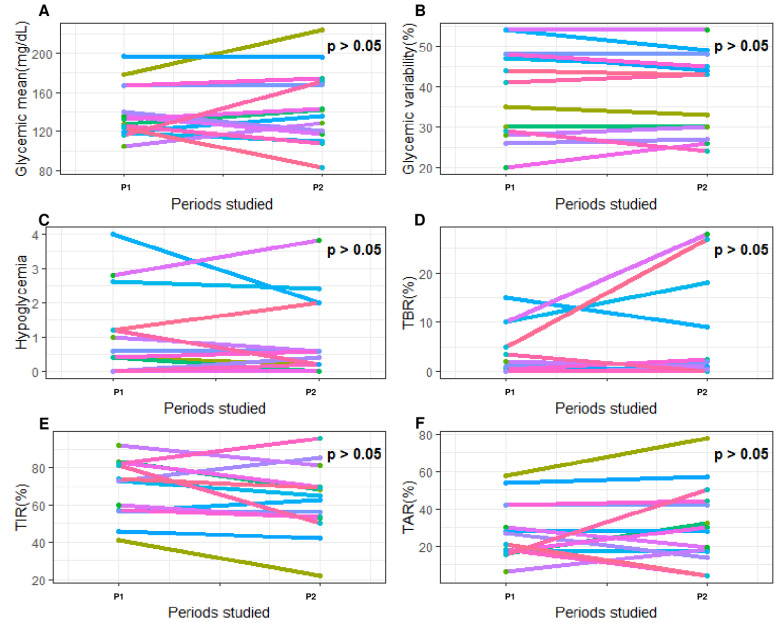
Evolution of glycemic parameters between two periods (P1–P2) of 5 days separated by 20 days without any severe acute complication related to type 1 diabetes. The *x*-axis represents the 2 periods studied around severe hypoglycemia. P1, the first 5-day period; P2, the second 5-day period. The *y*-axis corresponds to the glycemic parameters: (**A**) glycemic mean (mg/dL); (**B**) coefficient of variability (%); (**C**) frequency of hypoglycemia (n/day); (**D**) time spent affected by hypoglycemia (<3.3 mmol/L; %); (**E**) time spent affected by hyperglycemia (>8.8 mmol/L; %); (**F**) time in range (3.3–8.8 mmol/L; %). Each color of the graph corresponds to a single patient. On all graphs, *p*-values are written. Each *p*-value represents the statistical significance of the variation in the studied blood glucose parameter between the two periods concerned. Differences between two periods were considered as significant when *p*-value was under 0.05.

**Table 1 nutrients-15-02957-t001:** Description of cohorts.

	Total (*n* = 441)	Non-SH (*n* = 279)	SH (*n* = 162)	*p*-Value
Clinical parameters				
Gender—Girls %	45	48	41	0.16 ‡
Age—years	13.65 ± 3.8	13.78 ± 3.8	13.44 ± 3.7	0.18 †
Boys	14 ± 3.8	13.98 ± 3.8	14.02 ± 3.8	0.47 †
Girls	13.2 ± 3.7	13.55 ± 3.8	12.62 ± 3.5	0.04 * †
Diabetes history				
Diabetes duration—years	5.91 ± 4.4	5.66 ± 4.2	6.32 ± 4.9	0.07 †
Boys	6.3 ± 4.8	5.92 ± 4.4	6.81 ± 5.3	0.08 †
Girls	5.5 ± 3.9	5.35 ± 3.9	5.62 ± 3.9	0.3 †
Insulin regime				
2 injections—%	10.4	9.7	11.7	0.46 ‡
Boys	7.1	7.6	6.3	0.69 ‡
Girls	14.5	11.2	19.7	0.14 ‡
Basal-prandial—%	68	65.6	72.2	0.15 ‡
Boys	67.6	66.9	69.8	0.64 ‡
Girls	68	64.2	75.8	0.17 ‡
Pump—%	21.6	24.7	16.1	0.03 * ‡
Boys	25.3	26.2	23.9	0.69 ‡
Girls	17.5	23.9	4.5	7 × 10^−4^ *** ‡
Partial remission				
Frequency—%	62	62.6	60.5	0.82 ‡
Duration—months	5.78 ± 6.4	5.62 ± 6.4	6.03 ± 6.4	0.33 †
In process—%	33	31.3	36.7	0.6 ‡
Glycemic parameters				
Glycemic mean—mmol/L	9.21 ± 2.07	9.34 ± 2.16	8.97 ± 1.89	0.02 * †
Glycated hemoglobin—%	7.7 ± 1.4	8.0 ± 1.47	7.24 ± 1.19	1 × 10^−7^ *** †
Insulino-resistance—UI/kg/day	0.9 ± 0.3	0.94 ± 0.31	0.94 ± 0.35	0.5 †
TBR_<3.3_—%	10.5 ± 7.5	8.3 ± 6.4	14.3 ± 7.6	1 × 10^−6^ *** †
TIR_3.3–8.8_—%	47.1 ± 15	49 ± 15.1	43.7 ± 14.2	1 × 10^−4^ *** †
TAR_>8.8_—%	42.5 ± 15.7	42.3 ± 16.4	42.4 ± 15.4	0.47 †

Plus–minus values are means ± SD. Percentages may not total 100 due to rounding. Differences between SH and non-SH were considered as significant when *p*-value was under 0.05. The level of significance is represented as follows: nonsignificant (), *p* < 0.05 (*), *p* < 0.001 (***). † Student *t*-test; ‡ Chi-square. SH, medical consultation following severe hypoglycemia; non-SH, medical consultation not preceded by a severe hypoglycemia or ketoacidosis; in process, patients with de novo diabetes in partial remission at the time of consultation; TIR_3.3–8.8_, time in range (3.3–8.8 mmol/L); TBR_<3.3_, time below the range (<3.3 mmol/L); TAR_>8.8_, time above the range (>8.8 mmol/L).

**Table 2 nutrients-15-02957-t002:** Specific threshold to anticipate severe hypoglycemia.

Parameters	Cut-Off (% Change)	Sensitivity (%)	Specificity (%)	PPV (%)	NPV (%)
Glycemic mean	−10	50	74	67	58
Hypoglycemia frequency	10	81	60	68	75
TAR_>8.8_	50	75	53	63	67
TBR_<3.3_	−5	56	73	69	61

The intensity of the variation in the glycemic parameters is calculated using the following formulas ([P2–P1]/P1) or ([H2–H1]/H1). The cut-offs were determined from ROC curves. Abbreviations: TBR_<3.3_, time below the range (<3.3 mmol/L); TAR_>8.8_, time above the range (>8.8 mmol/L); PPV, positive predictive value; NPV, negative predictive value.

## Data Availability

The data presented in this study are available in the article.

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
