# Peer review of "Deep Analysis of Clinical Parameters and Temporal Evolution of Glycemic Parameters Based on CGM Data for the Characterization of Severe Hypoglycemia in a Cohort of Children and Adolescents with Type 1 Diabetes"

_nutrients, 2023, doi:10.3390/nu15132957_

Round 1
Reviewer 1 Report
See Letter attached,
Title:
Deep analysis of clinical parameters and temporal evolution of glycemic parameters based on CGM data for the characterization of severe hypoglycemia in a cohort of children and adolescents with type 1 diabetes.
Reviewer's comments
The intent of this study is laudable. There have been many published studies that developed models to predict hypoglycemia among people with diabetes The recent literature consists mostly of studies using machine learning algorithms to analyse "big data" derived from large patient databases that included Continuous Glucose Monitoring records [Zhang L, et al. Data-based modeling for hypoglycemia prediction: Importance, trends, and implications for clinical practice. Front Public Health. 2023; 11: 1044059. doi: 10.3389/fpubh.2023.1044059.] This study does not meet the current research standard but I do not dismiss it entirely, because it is original research and it attempts to include CGM data; therefore this study represents an improvement over the body of earlier literature describing prediction models based on clinical and laboratory parameters.
Methodological deficiencies:
1. How many patient visits had CGM profiles sufficient to allow calculation of the glycemic parameters (TBR<60, TIR60-160, TAR>160)? Was it 441, or 42, or some number in between? The manuscript contains contradictory statements on this critical matter. If it was only 42, this challenges the assumptions of sensitivity and specificity, the validity of the ROC curve analysis, and the reliability of the subsequent estimates of positive and negative predictive value.
2. Figure 2 and the accompanying text do not sufficiently explain how you arrived at the supposedly optimal TBR<60 cut-off of 9%. Show us the Receiver Operating Characteristic curve with sensitivity on the vertical axis and 1-specificity on the horizontal axis. Then readers can see the trade-off between sensitivity and specificity as one varies the cut-off. There are various ways of defining "optimal" in ROC curve analysis, but it is not obvious that a cut-off of 9% (sensitivity 71% and specificity 63%) is optimal. You say that a cut-off of 10% will give sensitivity of 80% and specificity of 60%, which seems better, if one wanted to maximize sensitivity+specificity.
3. The analysis of TBR<60 as a predictor of severe hypoglycemia is incomplete and does not lead to any useful conclusion. As shown in the 2x2 table analysis below, TBR<60 performs rather poorly. Assuming that sensitivity = 0.71 and specificity = 0.63, in your study cohort, where the prevalence of SH is 162/441 = 0.367, if the test is "positive" (i.e., TBR<60 more than 9% of the time) the estimated probability of SH (the positive predictive value) would be 0.527 (95% CI: 0.461 to 0.593). If the test is "negative" (i.e., TBR<60 less than 9% of the time) the estimated probability of SH (1 minus the negative predictive value) would be 0.211 (95% CI: 0.157 to 0.264). I suggest that these estimated probabilities are not sufficiently different from the pre-test prevalence to make the test practically useful as a decision criterion. I assumed that you had 441 TBR<60 observations. If you had only 42, then the subsequent confidence intervals for the estimated predictive values would be so wide as to make the study findings truly useless.
|
SH + |
SH - |
Total |
|
|
|
TBR + |
115 |
103 |
218 |
|
|
|
TBR - |
47 |
176 |
223 |
|
|
|
Total |
162 |
279 |
441 |
|
|
|
|
|
|
|
|
|
|
prevalence of SH |
0.367 |
|
|
|
L95CL |
U95CL |
TBR cut-off |
0.090 |
|
PPV |
0.527 |
0.461 |
0.593 |
sensitivity |
0.710 |
|
NPV |
0.789 |
0.736 |
0.843 |
specificity |
0.630 |
|
1-NPV |
0.211 |
0.157 |
0.264 |
The predictive values could be improved by developing a composite risk score. In Table 1 you identified five variables as being statistically significantly associated with SH: TBR<60 (%), TIR60-160 (%), use of insulin pump (yes/no), glycemic mean, and HbA1c. How did you decide that the prediction model should include only TBR<60 as the hypothesized predictor? You need to test for confounding, independent effects, and interactions with the other variables: I advise multivariable logistic regression analysis. For purposes of multivariable analysis, keep TBR<60 (%), TIR60-160 (%), glycemic mean, and HbA1c as interval-ratio scale variables, in order to maximize statistical power. Use of insulin pump remains a binary variable (1=yes, 0=no) as it can't be coded any other way. The best-fitting multivariable logistic regression model can then be used as a calculator to predict risk of SH. At this point, the risk estimated by the model can be dichotomized with a cut-off threshold, and you can use 2x2 table and ROC curve analysis to optimize the cut-off threshold and evaluate the performance of the risk prediction model. Certainly you will thus find a combination of variables that will predict SH better than TBR<60 alone.
Major editing required:
4. As this is a journal aimed at an international audience, I advise converting United States (mg/dL) blood glucose measurements to Système International units (mmol/L).
Introduction:
5. L68: Replace "letting" with "leaving". Replace "diagnose" with "identify".
6. L69-71: Replace the statement of objectives sentence with this: "The objective of this analysis to identify glycemic and clinical markers of risk for SH in children and adolescents with type 1 diabetes."
Methodology:
7. L85: Before "presence" insert "and".
8. L87-97: Before each of the listed exclusion criteria insert "or".
9. L116: Since "Severe Hypoglycemia" is the primary outcome variable, insert a clear statements of your definition of SH and how you ascertained its occurrence. I presume this would be partly determined by clinical interview (How did you expect patients to report and describe "altered consciousness"?) and partly based on corroborating glucose measurements (how low and for how long?). In what time period prior to the visit did the SH event(s) occur? How many episodes were needed to meet the definition?
10. L116: Definitions of the other glycemic variables (TBR<60, TIR60-160, TAR>160) need to be inserted here.
11. L123: Delete the statements about linear regression, ANOVA, Kruskal-Wallis tests, and Bonferroni procedure. I don't see that you used such methods in this study. Was this section cut and pasted from another paper?
12. L131: On what basis did you select the 441 outpatient visits from among the more than 5,000 that met the previously stated inclusion criteria? I presume this had something to do with the availability of outpatient Continuous Glucose Monitoring data. How complete, or how prolonged did the CGM need to be to be included? Please explain. Also, how many were excluded because of medical complications? Give us a list of all reasons for exclusion and the numbers excluded.
13. The secondary analysis of the evolution of severe hypoglycemic events was based on the CGM records of 42 patients (L179). Explain how these 42 were selected from among 441.
Results:
14. Table 1:
Include three columns with the "N" (number of observations among the total cohort, the non-SH group, and the SH group) for each of the listed parameters. The totals in the header can't be taken as the N for every parameter. There must have been some missing observations. Also, this would clarify if the glycemic parameters were observed in 42 subjects or 441.
Use consistent abbreviations. In the header, replace "non-Hypo group" with "non-SH", and replace "Hypo-group" with "SH".
Replace "Chronic treatment" with "Insulin regime".
Footnote, L157: Replace "severe acute complication" with "severe hypoglycemia or ketoacidosis in previous 12 months".
15. L179-291: The secondary analysis of the evolution of severe hypoglycemic events (L179-291) is very long and very confusing. What is the purpose, and of what significance are the findings? I advise deleting this part of the report.
16. L293-301: This part, and the following Table 2 are incomprehensible. Was the ROC analysis based on 42 CGM records or 441? Show the ROC curves and explain how you arrived at the supposedly optimal cut-off thresholds. Some of the stated cut-offs are impossible: how can a cut-off % be less than zero? State the prevalence of SH among the 42 subjects (or was it 441?) Given the specified sensitivity and specificity levels, the stated PPVs and NPVs are improbable: they do not agree with my own 2x2 table calculations.
17. Table 2, column headings: Replace "VPP" with "PPV". Replace "VPN" with "NPV".
Discussion:
18. L324: As shown in Table 1, you did find one clinical parameter statistically significantly associated with lower probability of SH: use of an insulin pump. Indeed, that is considered one of the advantages of using this device.
19. L328-331: You are grossly overstating the value of TBR<60 alone as a predictor of SH, and as a guideline for revision of insulin management. If you develop a better prediction model, you might have grounds for making such recommendations.

English language use is satisfactory. A few minor edits are needed. (see letter).
Author Response
Open Review
See Letter attached,
Title:
Deep analysis of clinical parameters and temporal evolution of glycemic parameters based on CGM data for the characterization of severe hypoglycemia in a cohort of children and adolescents with type 1 diabetes.
Reviewer's comments
The intent of this study is laudable. There have been many published studies that developed models to predict hypoglycemia among people with diabetes The recent literature consists mostly of studies using machine learning algorithms to analyse "big data" derived from large patient databases that included Continuous Glucose Monitoring records [Zhang L, et al. Data-based modeling for hypoglycemia prediction: Importance, trends, and implications for clinical practice. Front Public Health. 2023; 11: 1044059. doi: 10.3389/fpubh.2023.1044059.] This study does not meet the current research standard but I do not dismiss it entirely, because it is original research and it attempts to include CGM data; therefore this study represents an improvement over the body of earlier literature describing prediction models based on clinical and laboratory parameters.
We thank the reviewer for the very useful comments.
Methodological deficiencies:
- How many patient visits had CGM profiles sufficient to allow calculation of the glycemic parameters (TBR<60, TIR60-160, TAR>160)? Was it 441, or 42, or some number in between? The manuscript contains contradictory statements on this critical matter. If it was only 42, this challenges the assumptions of sensitivity and specificity, the validity of the ROC curve analysis, and the reliability of the subsequent estimates of positive and negative predictive value.
Our study was retrospective and divided into two parts. The first part analyzed the data collected during each outpatient visit (or consultation) by our diabetes nurses. To collect this data, the nurses retrieved data recorded on the patients' subcutaneous glucose sensors. To be encoded in the medical consultation note, patients needed to have collected at least 50% of glucose values. These data only represented the summary of blood glucose levels prior to severe hypoglycemia. Statistical analyses were carried out on data collected during 441 medical consultations.
The second part of the study focused on CGM data from patients who have experienced severe hypoglycemia. To analyze these CGMs, the patient had to be registered on the Libreview platform. Unfortunately, only 42 patients were registered on Libreview. Statistical analyses were performed on data collected from the 42 CGMs.
- Figure 2 and the accompanying text do not sufficiently explain how you arrived at the supposedly optimal TBR<60 cut-off of 9%. Show us the Receiver Operating Characteristic curve with sensitivity on the vertical axis and 1-specificity on the horizontal axis. Then readers can see the trade-off between sensitivity and specificity as one varies the cut-off. There are various ways of defining "optimal" in ROC curve analysis, but it is not obvious that a cut-off of 9% (sensitivity 71% and specificity 63%) is optimal. You say that a cut-off of 10% will give sensitivity of 80% and specificity of 60%, which seems better, if one wanted to maximize sensitivity+specificity.
In the revised version of the manuscript, Figure 2 was replaced by the desired ROC curve.
- The analysis of TBR<60 as a predictor of severe hypoglycemia is incomplete and does not lead to any useful conclusion. As shown in the 2x2 table analysis below, TBR<60 performs rather poorly. Assuming that sensitivity = 0.71 and specificity = 0.63, in your study cohort, where the prevalence of SH is 162/441 = 0.367, if the test is "positive" (i.e., TBR<60 more than 9% of the time) the estimated probability of SH (the positive predictive value) would be 0.527 (95% CI: 0.461 to 0.593). If the test is "negative" (i.e., TBR<60 less than 9% of the time) the estimated probability of SH (1 minus the negative predictive value) would be 0.211 (95% CI: 0.157 to 0.264). I suggest that these estimated probabilities are not sufficiently different from the pre-test prevalence to make the test practically useful as a decision criterion. I assumed that you had 441 TBR<60 observations. If you had only 42, then the subsequent confidence intervals for the estimated predictive values would be so wide as to make the study findings truly useless.
|
SH + |
SH - |
Total |
|
|
|
TBR + |
115 |
103 |
218 |
|
|
|
TBR - |
47 |
176 |
223 |
|
|
|
Total |
162 |
279 |
441 |
|
|
|
|
|
|
|
|
|
|
prevalence of SH |
0.367 |
|
|
|
L95CL |
U95CL |
TBR cut-off |
0.090 |
|
PPV |
0.527 |
0.461 |
0.593 |
sensitivity |
0.710 |
|
NPV |
0.789 |
0.736 |
0.843 |
specificity |
0.630 |
|
1-NPV |
0.211 |
0.157 |
0.264 |
The predictive values could be improved by developing a composite risk score. In Table 1 you identified five variables as being statistically significantly associated with SH: TBR<60 (%), TIR60-160 (%), use of insulin pump (yes/no), glycemic mean, and HbA1c. How did you decide that the prediction model should include only TBR<60 as the hypothesized predictor? You need to test for confounding, independent effects, and interactions with the other variables: I advise multivariable logistic regression analysis. For purposes of multivariable analysis, keep TBR<60 (%), TIR60-160 (%), glycemic mean, and HbA1c as interval-ratio scale variables, in order to maximize statistical power. Use of insulin pump remains a binary variable (1=yes, 0=no) as it can't be coded any other way. The best-fitting multivariable logistic regression model can then be used as a calculator to predict risk of SH. At this point, the risk estimated by the model can be dichotomized with a cut-off threshold, and you can use 2x2 table and ROC curve analysis to optimize the cut-off threshold and evaluate the performance of the risk prediction model. Certainly you will thus find a combination of variables that will predict SH better than TBR<60 alone.
First of all, let us express my gratitude for your constructive comments.
Let us please recapitulate the main objective of our statistical analysis: to find a reliable and practical parameter to reduce the risk of complications. It is essential that this parameter can be easily used in clinical practice, without the need for complex calculations, to provide a clear and precise therapeutic solution for patients or primary caregivers who do not necessarily have in-depth medical insight in diabetology.
To select the appropriate clinical parameter, we performed a preliminary logistic regression to determine which parameter was most reliable in predicting severe hypoglycemia. The first three parameters considered were mean glucose (Pr < 2e-16), glycated hemoglobin (Pr < 2e-16) and TBR (Pr = 6.13e-6). We then performed a multi-variate analysis. With this method, we obtained an equation that was too complex to use in clinical practice. We opted for a precise parameter to facilitate clinical management.
However, the first two parameters presented a major disadvantage. Increasing mean blood glucose and glycated hemoglobin could reduce the risk of severe hypoglycemia, but it could also increase the risk of long-term chronic complications. We have therefore chosen to focus on reducing the TBR. Furthermore, our analysis showed that a TBR of less than 9% gave a relative risk of severe hypoglycemia of 0.42 and a relative risk reduction of 60%.
It's important to note that reducing the TBR would also have an impact on mean blood glucose and glycated hemoglobin. In choosing this parameter, we took into account not only its effectiveness in reducing the risk of severe hypoglycemia, but also its ability to positively influence other relevant clinical measures.
This information has been added to the revised text. (Line - 185 and line - 352)
Major editing required:
- As this is a journal aimed at an international audience, I advise converting United States (mg/dL) blood glucose measurements to System International units (mmol/L).
All changes were made in the text of our revised version of the manuscript.
Introduction:
- L68: Replace "letting" with "leaving". Replace "diagnose" with "identify".
All changes were made in the text of our revised version of the manuscript.
- L69-71: Replace the statement of objectives sentence with this: "The objective of this analysis to identify glycemic and clinical markers of risk for SH in children and adolescents with type 1 diabetes."
All changes were made in the text of our revised version of the manuscript.
Methodology:
- L85: Before "presence" insert "and".
All changes were made in the text of our revised version of the manuscript.
- L87-97: Before each of the listed exclusion criteria insert "or".
All changes were made in the text of our revised version of the manuscript.
- L116: Since "Severe Hypoglycemia" is the primary outcome variable, insert a clear statements of your definition of SH and how you ascertained its occurrence. I presume this would be partly determined by clinical interview (How did you expect patients to report and describe "altered consciousness"?) and partly based on corroborating glucose measurements (how low and for how long?). In what time period prior to the visit did the SH event(s) occur? How many episodes were needed to meet the definition?
In the revised version of our manuscript, a paragraph has been added to the methodology to define severe hypoglycemia. Two groups: one (severe hypoglycemia group or SH group) composed of all consultations (n=162) that followed a recent SH episode (less than 3 weeks between SH and consultation; mean=17 days)”. This information has been added to the revised text. (Line – 109)
- L116: Definitions of the other glycemic variables (TBR<60, TIR60-160, TAR>160) need to be inserted here.
Definitions were added in the text of our revised version of the manuscript. (Line - 128)
- L123: Delete the statements about linear regression, ANOVA, Kruskal-Wallis tests, and Bonferroni procedure. I don't see that you used such methods in this study. Was this section cut and pasted from another paper?
The Bonferroni correction was used to analyze glycemic patterns in the second part of the study. Analyses verified by a statistician.
- L131: On what basis did you select the 441 outpatient visits from among the more than 5,000 that met the previously stated inclusion criteria? I presume this had something to do with the availability of outpatient Continuous Glucose Monitoring data. How complete, or how prolonged did the CGM need to be to be included? Please explain. Also, how many were excluded because of medical complications? Give us a list of all reasons for exclusion and the numbers excluded.
The SH group included the 162 severe hypoglycemia experienced by our patients during the analysis period.
So 162 consultations.
The non-SH group were based on one medical consultation for each of our 279 patients enrolled in the Cliniques universitaires Saint-Luc pediatric diabetology convention.
So 279 patients
This gives us a total of 441 patients.
Incomplete consultation files were not analyzed (absence of precise blood glucose measurement, absence of clinical parameters analyzed, etc.). Other exclusion criteria are listed in the methodology.
- The secondary analysis of the evolution of severe hypoglycemic events was based on the CGM records of 42 patients (L179). Explain how these 42 were selected from among 441.
Please refer to our explanation mentioned above.
As a reminder,
Our study was retrospective and divided into two parts. The first part analyzed the data collected during each outpatient visit (or consultation) by our diabetes nurses. To collect this data, the nurses retrieved data recorded on the patients' subcutaneous glucose sensors. To be encoded in the medical consultation note, patients needed to have collected at least 50% of glucose values. These data only represented the summary of blood glucose levels prior to severe hypoglycemia. Statistical analyses were carried out on data collected during 441 medical consultations.
The second part of the study focused on CGM data from patients who have experienced severe hypoglycemia. To analyze these CGMs, the patient had to be registered on the Libreview platform. Unfortunately, only 42 patients were registered on Libreview. Statistical analyses were performed on data collected from the 42 CGMs.
Results:
- Table 1:
Include three columns with the "N" (number of observations among the total cohort, the non-SH group, and the SH group) for each of the listed parameters. The totals in the header can't be taken as the N for every parameter. There must have been some missing observations. Also, this would clarify if the glycemic parameters were observed in 42 subjects or 441.
Data were collected from 441 complete medical records. As described above, incomplete files were not selected.
Use consistent abbreviations. In the header, replace "non-Hypo group" with "non-SH", and replace "Hypo-group" with "SH".
All changes were made in the text of our revised version of the manuscript.
Replace "Chronic treatment" with "Insulin regime".
All changes were made in the text of our revised version of the manuscript.
Footnote, L157: Replace "severe acute complication" with "severe hypoglycemia or ketoacidosis in previous 12 months".
All changes were made in the text of our revised version of the manuscript.
- L179-291: The secondary analysis of the evolution of severe hypoglycemic events (L179-291) is very long and very confusing. What is the purpose, and of what significance are the findings? I advise deleting this part of the report.
The aim of this section was to evaluate more precisely the evolution of glycemic parameters in the days preceding severe hypoglycemia. These analyses showed that hypoglycemia was not an isolated event, but led to changes in all glycemic parameters in the days before and after the incident. Unfortunately, these parameters did not evolve in a specific pattern that would have made it possible to anticipate this severe hypoglycemia.
- L293-301: This part, and the following Table 2 are incomprehensible. Was the ROC analysis based on 42 CGM records or 441? Show the ROC curves and explain how you arrived at the supposedly optimal cut-off thresholds. Some of the stated cut-offs are impossible: how can a cut-off % be less than zero? State the prevalence of SH among the 42 subjects (or was it 441?) Given the specified sensitivity and specificity levels, the stated PPVs and NPVs are improbable: they do not agree with my own 2x2 table calculations.
Cut-offs are calculated on the basis of variations in the various glycemic parameters studied. For example, the frequency of hypoglycemia increased in the days preceding a severe hypoglycemia compared with a period 10% further away, according to our analyses. On the other hand, we have shown that mean blood glucose levels tended to fall in the days preceding hypoglycemia. According to our calculations, mean blood glucose levels decreased by 10%. It is therefore normal to have positive or negative cut-offs with the formula we have used.
- Table 2, column headings: Replace "VPP" with "PPV". Replace "VPN" with "NPV".
All changes were made in the text of our revised version of the manuscript.
Discussion:
- L324: As shown in Table 1, you did find one clinical parameter statistically significantly associated with lower probability of SH: use of an insulin pump. Indeed, that is considered one of the advantages of using this device.
We thank the reviewer for this comment.
- L328-331: You are grossly overstating the value of TBR<60 alone as a predictor of SH, and as a guideline for revision of insulin management. If you develop a better prediction model, you might have grounds for making such recommendations.
Our present aim was indeed not to propose a modification of therapeutic guidelines, but simply to offer a tool for optimizing the risk of severe hypoglycemia. At present, there is no clinical parameter in the recommendations to guide diabetes management in the face of severe hypoglycemia. The advantage of the TBR is that it is easy to analyze in a 20-minute medical consultation and enables advice to be given to patients.
Comments on the Quality of English Language
English language use is satisfactory. A few minor edits are needed. (see letter).
Submission Date
06 June 2023
Date of this review
14 Jun 2023 15:53:08

Reviewer 2 Report
This is a well written paper in which data of the EPI-GLUREDIA-study are reviewed retrospectively. The paper provides considerable insight to identify clinical and glycemic parameters to refine the prediction, prevention and treatment of patients at risk for SH. The patho-physiology is well described and illustrated with trial data/graphical displays (e.g. Figures 5 -7). The added value of the paper is the observation of a relative young age group where prevention of the consequences of T1D/SH can play a significant role later in life. As is, the paper reads well and the data are well presented, also graphically (in particular Fig. 2).
Minor point:
Table 1. Please write TBR, TIR and TAR in full in the legenda of the table.
The same suggestion to systematically review the manuscript text and legenda of the tables and figures on abbreviations. Some abbreviations may seem obvious for those specialized in diabetes (for example CGM = Continuous Glucose Monitoring), but as the paper is of interest to other disciplines (preventive cardiology, vascular medicine etc.) as well, writing all abbreviations in full with every Figure and table as well as throughout the text helps the non-specialist reader.
Figure 2. Very elegant figure with the density curves. The term 'density' may not be known to all readers. I suggest the authors add 'density = '(data) frequency distribution'. The first sentence of the legendum then reads 'Density (or frequency distribution) curves of patients per group (SH-group vs non-SH-group) according to their time spent 173 in hypoglycemia. Just as a suggestion.
Reviewer 3 Report
In this study, Harvengt et al., found that Time below the range<60 <9% is a sensitive and specific tool to reduce the frequency of severe hypoglycemia. Although the study was well-designed, there still several issues should be addressed.
1. In the introduction section, the clinical significance should be emphasized to increase the interests of the readers.
2. The quality of each figure should be improved.
Round 2
Reviewer 1 Report
The responses to my previous criticisms ought better have been incorporated into the text of the manuscript. However, if my comments and the authors' responses are published together as an open review, then this is acceptable.
Minor editing still needed:
L48: Replace "8.25 mmol/L" with "8.33 mmol/L"
Figure 1, footnote: After "162 consultations preceded by severe hypoglycemia" insert "in the past 3 weeks".
Table 1: Replace "insulin regim" with "insulin regime".
Table 2: Replace "Cut-off (%)" with "Cut-off (% change)".